# The association of different types of physical activity and diabetes co-morbid depression: A cross-sectional analysis

**Yage Yang[1‡], Hongzhen Liu[2]\***

**1** Zhengzhou Sias University, Zhengzhou, Henan, People's Republic of China, **2** School of Physical Education (Main Campus), Zhengzhou University, Zhengzhou, Henan, People's Republic of China

‡ These authors share first authorship on this work.
* Zzdxliuhongzhen@163.com

## Abstract

### Background

Diabetes co-morbid depression is a significant public health burden. Physical activity (PA) has been suggested as a potential approach to reduce the risk of diabetes co-morbid depression. Different types of PA may have different effects on diabetes co-morbid depression.

### Objective

The aims of this study were to investigate the association between moderate to vigorous physical activity (MVPA), different types of physical activity, including work activity (WPA), transportation physical activity (TPA), recreational physical activity (RPA), sedentary behavior (SB), and co-morbid depression in participants with diabetes.

### Materials and methods

The data for this study were derived from the 2017-2018 National Health and Nutrition Examination Survey (NHANES). A total of 642 participants aged 20 years and above were included in the study (mean age: 63.54 ± 12.08 years; 367 males and 275 females). Depression was screened by PHQ-9 in participants who were told to have diabetes by a doctor. PAs were screened by GPAQ. A binary logistic regression model was performed to analyze the association of RPA and diabetes co-morbid depression.

### Results

The causal relationship between MVPA and diabetes co-morbid depression did not reach a significant level (*P*=0.949), nor did it reach in WPA (*P*=0.203), TPA (*P*=0.299) and SB (*P*=0.219). RPA had a significant effect on diabetes co-morbid depression

**Data availability statement:** The datasets generated and/or analyzed during the current study are available in the [NHANES] repository, [NHANES Questionnaires, Datasets, and Related Documentation (cdc.gov)]. Raw data supporting the obtained results are available at the corresponding author.

**Funding:** The author(s) received no specific funding for this work.

**Competing interests:** The authors have declared that no competing interests exist.

**Abbreviations:** PA: Physical activity; MVPA: moderate to vigorous physical activity; WPA: work physical activity; TPA: transportation physical activity; RPA: recreational physical activity; SB: sedentary behavior; NHANES: National Health and Nutritional Examination Survey; GPAQ: Global Physical Activity Questionnaire; PHQ-9: The Patient Health Questionnaire; BMI: Body Mass Index; OR: Odds Ratio; CI: Confidence Interval; P: P-Value; U.S.: The United States of America; MEC: Mobile Examination Center; CAPI: Computer-Assisted Personal Interview; CDC: Centers for Disease Control and Prevention; NCHS: National Center for Health Statistics; CRF: Corticotropin Releasing Factor; SPSS: Statistical Package for Social Sciences.

(OR=0.508, 95%CI: 0.347-0.742, $P<0.001$), the effect remained significant after adjusted for confounding variables (OR=0.522, 95%CI: 0.356-0.789, $P$=0.002).

## Conclusions

Among the various types of physical activity, only RPA was a protective factor for co-morbid depression in diabetes.

---

## 1 Background

Diabetes co-morbid depression is an important public health burden as it has been linked to poorer diabetes self-care, increased risk of diabetes complications and mortality, as well as a higher burden on healthcare and self-independent. According to a meta-analysis of 42 studies, individuals with diabetes have a higher prevalence of depression compared to those without diabetes [1]. Finding effective preventive measures to prevent depression in people with diabetes is essential to improve their physical and mental health.

Physical activity (PA) has been suggested as a potential approach to reduce the risk of diabetes co-morbid depression. PAs have been shown to improve glycemic control, insulin sensitivity, blood pressure, and lipid profiles, as well as have positive effects on mood and psychological well-beings [2]. A meta-analysis of 32 randomized controlled trials showed that PA interventions were effective in reducing depressive symptoms among adults with diabetes [3]. Furthermore, a longitudinal cohort study also suggested that higher levels of physical activity were associated with a reduced risk of developing depression in individuals with type II diabetes [4]. Therefore, PA may be a promising non-pharmacological intervention or prevention method to prevent or manage depression in patients with diabetes.

Different types of PA may have different effects on diabetes co-morbid depression. According to the Global Physical Activity Scale (GPAQ), PA can be divided into four dimensions: Work physical activity (WPA), transportation physical activity (TPA), recreational physical activity (RPA), and sedentary activity. A meta-analysis of 19 studies found that higher levels of WPA were associated with a reduced risk of depression in adults, including those with diabetes [5]. A prospective cohort study of Japanese adults found that the prevalence rate of depression lowered significantly among the participants with diabetes who engaged in walking intervention [6]. Furthermore, a randomized control trial also found that higher levels of recreational physical activity were associated with a reduced risk of developing depression in individuals with type 2 diabetes [7]. As for sedentary activity, a meta-analysis of 19 studies found that sedentary behavior (SB) was positively associated with depressive symptoms in adults [8]. Another meta-analysis showed that reducing sedentary behavior may have a beneficial effect on depressive symptoms [9]. In PAs with a certain intensity, the results were mostly positive. However, in addition to these positive findings, we believe that different types of physical activity may have different degrees of impact on the risk of depression in

co-morbid diabetes and that the selection of appropriate physical activity interventions may be important for achieving optimal mental health outcomes. The aims of this study were to investigate the association between moderate to vigorous physical activity (MVPA), different types of physical activity, including WPA, TPA, RPA, SB, and co-morbid depression in participants with diabetes.

## 2 Materials and methods

### Ethics approval and consent to participate

All procedures performed in the study were in accordance with the Declaration of Helsinki. The study protocols for NHANES were approved by the National Center for Health Statistics (NCHS) Research Ethics Review Board (Protocol#2017–1). All adult participants provided written notification of consent before participating in the study.

**2.1 Study population.** The National Health and Nutrition Examination Survey (NHANES) is a population-based cross-sectional survey designed to collect information about the health and nutrition situation of the US household population. A stratified multistage sampling design was used to obtain a representative sample of U.S. residents of two months or older; a two-year survey cycle covers about 15000 families. The NHANES protocol was approved by the National Center for Health Statistics (NCHS) research ethics review board; all adult participants provided written notice of consent [10]; the use of NHANES data as a secondary data source was also approved by the NCHS [11]. The present study extracted and aggregated data on diabetes, depression, physical activity, demographic characteristics and body measures data from the NHANES 2017–18 survey cycle, and the current sample is restricted to adults aged 20 or older with diabetes. After excluding missing and invalid data, a final total of 642 samples were included in this study. For more detailed information regarding the study design, sampling, and exclusion criteria, please refer to the figure below. (See Fig 1).

**2.2 Diabetes Co-morbid depression assessment.** In this study, we selected individual participants with both diabetes and depression as study subjects, only participants with diabetes were included in the study.

The NHANES 2017−18 "Diabetes (DIQ_J)" dataset was used in this analysis for diabetes. The diabetes section (variable name prefix DIQ) provides personal interview data on diabetes, prediabetes, use of insulin or oral hypoglycemic medications, and diabetic retinopathy. It also provides self-reported information on awareness of risk factors for diabetes, general knowledge of diabetic complications, and medical or personal cares associated with diabetes. Variable "DIQ010 - Doctor told you have diabetes" was used to access weather the participants have diabetes. This variable consisted of a question "Other than during pregnancy, ever been told by a doctor or health professional that you have diabetes or sugar diabetes?", and participants who answered "Yes" were defined as having diabetes.

The NHANES 2017–18 "Mental Health - Depression Screener (DPQ_J)" dataset was used in this analysis for depression. The Patient Health Questionnaire (PHQ-9), a nine-item depression screening instrument, was used to assess the frequency of depression symptoms in the sample over the past 2 weeks. For each item, points ranging from 0 to 3, are associated with the response categories "not at all", "several days", "more than half the days", and "nearly every day" [12,13]. A total score, ranging from 0 to 27, can be calculated for persons with complete responses to the symptom questions. In this study, PHQ-9 scores of 0–4 were classed as the "non-depression group' and scores of 5-27 were classed as the "depression group" [13].

**2.3 MVPA and different types of PA adjustments.** The NHANES 2017−18 "Physical Activity (PAQ_J)" dataset was used in this analysis for PAs. This questionnaire classifies physical activity into the following four categories: work-related physical activity, recreational physical activity, commuting physical activity, and sedentary behavior [14]. The adult section of this questionnaire encompasses items from PAQ605 to PAQ680. Designed based on the Global Physical Activity Questionnaire (GPAQ), it can provide interview data on various types of physical activity at the respondent level, enabling the assessment of whether the sample engaged in different types of physical activity during a typical week [15]. Participants aged 18 years and over were eligible for the adult section of PAs.

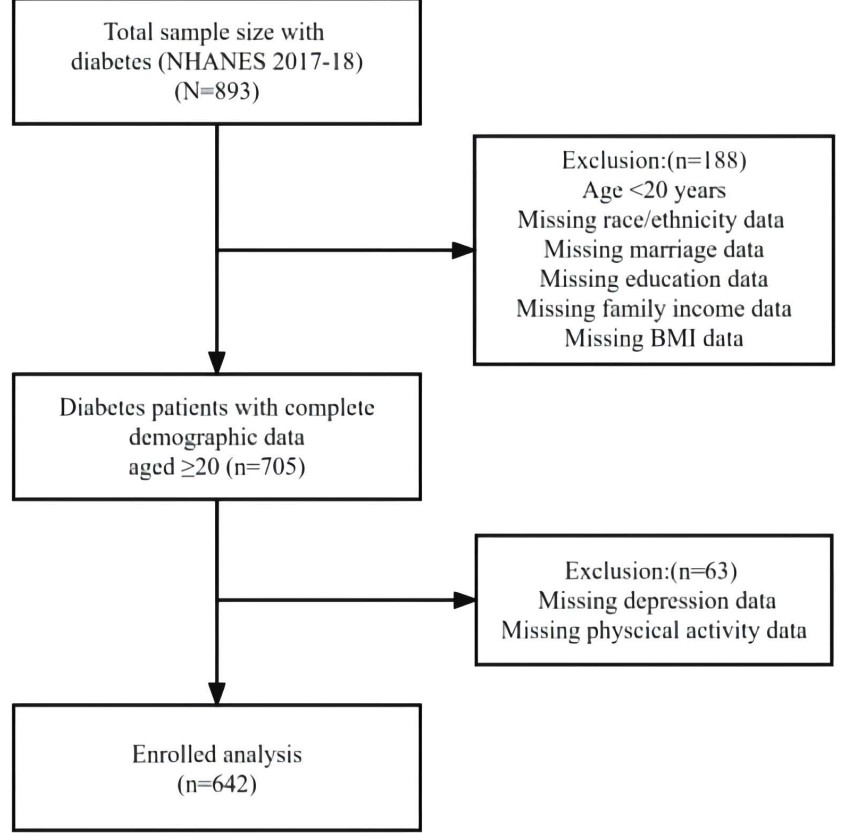

**Fig 1. Flow chart of subject selection.**

Variable "PAQ605 - Vigorous work activity" and variable "PAQ620 - Moderate work activity" was used to access weather the participants have moderate to vigorous WPA. This two variable consisted of question "Does your work involve vigorous-intensity activity that causes large increases in breathing or heart rate like carrying or lifting heavy loads, digging or construction work for at least 10 minutes continuously?" and "Does your work involve moderate-intensity activity that causes small increases in breathing or heart rate such as brisk walking or carrying light loads for at least 10 minutes continuously?" and participants who answered at least one "Yes" in these two questions were defined as having moderate to vigorous WPA [14,15].

Variable "PAQ635 - Walk or bicycle" was used to access weather the participants have TPA. This variable consisted of a question "In a typical week do you walk or use a bicycle for at least 10 minutes continuously to get to and from places?", and participants who answered "Yes" were defined as having TPA [14,15].

Variable "PAQ650 - Vigorous recreational activities" and variable "PAQ665 - Moderate recreational activities" was used to access weather the participants have moderate to vigorous RPA. This two variable consisted of question "In a typical week do you do any vigorous-intensity sports, fitness, or recreational activities that cause large increases in breathing or heart rate like running or basketball for at least 10 minutes continuously?" and "In a typical week do you do any moderate-intensity sports, fitness, or recreational activities that cause a small increase in breathing or heart rate such as brisk walking, bicycling, swimming, or volleyball for at least 10 minutes continuously?" and participants who answered at least one "Yes" in these two questions were defined as having moderate to vigorous RPA [14,15].

MVPA was also assessed by the above questions, and participants with at least one of WPA/TPA/RPA were defined as having MVPA. Variable "PAD680 - Minutes sedentary activity" was used to access weather the participants have SB. This variable consisted of a question "The following question is about sitting at school, at home, getting to and from places, or with friends including time spent sitting at a desk, traveling in a car or bus, reading, playing cards, watching television, or using a computer. Do not include time spent sleeping. How much time do you usually spend sitting on a typical day?", and participants report sedentary time in minutes. Sitting for 10 hours was well recognized as one of the critical points of sedentary duration time [16–18]. Stay sedentary for more than 10 hours could be devastating to human health. An authoritative review demonstrated that sedentary activity more than 10h/d had a significant negative impact on the human body, increasing the risk of chronic diseases as well as all-cause mortality [19]. Thus, we choose ">600min" as the cut-off point to define SB [14,15].

**2.4 Covariates.** Covariates of this study included gender, age, race/ethnicity, education, marital status, income to poverty and body mass index (BMI). Participants were categorized into three age groups: 20–39, 40–59, and >60. Race/ethnicity was categorized as Hispanic, non-Hispanic White, non-Hispanic Black, non-Hispanic Asian and Other. Education was categorized as Below high school, High school, and post high school. Marital status was categorized as Cohabiting, Married living alone (widowed, divorced, separated,) and Unmarried. Income to poverty was calculated by dividing family income by the poverty guidelines for the survey year. Poverty guidelines vary by family size and geographic location [20]. For this study, poverty to income ratio was used to create two categories of income status, impoverished (<1.3) and moderate income (≥1.3) [21]. During the MEC physical examination, weight and height were measured by trained technicians using standardized equipment, BMI was categorized into four standard categories: Underweight (≤18.9 kg/m$^2$), Normal weight (19.0-24.9 kg/m$^2$), Overweight (25.0-29.9 kg/m$^2$), and Obese (≥30.0 kg/m$^2$) [22].

## 2.5 Statistical analyses

We first transformed the raw data to.xlsl format by R software and use Microsoft Excel 2010 to excluded missing and useless (refused, not known) data and calculate the mean and standard deviation of the descriptive data. Adults aged 20 years and older with 1) diabetes, 2) complete information of depression, all PAs, BMI and all other demographic characteristics involved in this study were included in the database. We compared the differences between the two sets of categorical data using chi-square tests; the differences between the two sets of continuous variables were compared using rank sum tests. The comparison of the differences between groups for PAs as predictors was performed using post-hoc analysis. Binary Logistic regression analysis was performed to analyze the association between RPA and diabetes co-morbid depression. Variables that were statistically significant in the mono-factor analysis were included in the logistic stepwise regression analysis. A-entry = 0.05 and a-exit = 0.10 were used to select and exclude independent variables.

In analysis for the association of RPA and diabetes co-morbid depression, we confirmed RPA as the independent variable (0 = no, 1 = yes) and diabetes co-morbid depression (0 = non-depression, 1 = depression) as the dependent variable. In order to exclude the influences of confounding variables, the following models were developed: Model I: Original model, no adjustment of any variables. Model II: Adjusted for the independent variables in Model I plus gender. Model III: adjusted for variables in Model II plus marital status. Model IV: adjusted for variables in Model III plus income to poverty. Model V: adjusted for variables in Model IV plus BMI status. All data analysis were preformed using the Statistical Package for Social Sciences (SPSS) version 28.0, p-values less than 0.05 were considered statistically significant (two-sided test).

## 3 Results

### 3.1 Demographic characteristics

Mean age of participants in this study was 63.54 ± 12.08 years at the time of examination, including 367 males and 275 females, with mean PHQ-9 score 3.77. Significant statistical differences were observed ($P < 0.001$) in different gender, marital status, income to poverty and BMI between the depression group and the non-depression group. Differences were not observed in age ($P = 0.060$), race ($P = 0.143$) and education ($P = 0.129$). (See Table 1)

**Table 1. Demographic characteristics of NHANES 2017-18 diabetes adults aged ≥20 by depression.**

| Characteristics, n% | Sample Capacity N=642 | | Depression n=195 | | Non-Depression n=447 | | Test statistics | P |
|---|---|---|---|---|---|---|---|---|
| **Gender** | | | | | | | 19.516[a] | <0.001*** |
| Male | 367 | (57.1) | 86 | (44.1) | 281 | (62.8) | | |
| Female | 275 | (42.8) | 109 | (55.8) | 166 | (37.1) | | |
| **Age Group** | | | | | | | −1.884[b] | 0.060 |
| 20-39 | 28 | (4.3) | 7 | (3.5) | 21 | (4.6) | | |
| 40-59 | 169 | (26.3) | 64 | (32.8) | 105 | (23.4) | | |
| ≥60 | 445 | (69.3) | 124 | (63.5) | 321 | (71.8) | | |
| **Race** | | | | | | | 6.860[a] | 0.143 |
| Hispanic | 144 | (22.4) | 38 | (19.4) | 106 | (23.7) | | |
| Non-Hispanic White | 239 | (37.2) | 83 | (42.5) | 156 | (34.8) | | |
| Non-Hispanic Black | 145 | (22.5) | 46 | (23.5) | 99 | (22.1) | | |
| Non-Hispanic Asian | 81 | (12.6) | 17 | (8.7) | 64 | (14.3) | | |
| Other | 33 | (5.1) | 11 | (5.6) | 22 | (4.9) | | |
| **Education** | | | | | | | 4.092[a] | 0.129 |
| Below high school | 158 | (24.6) | 50 | (25.6) | 108 | (24.1) | | |
| High school | 162 | (25.2) | 58 | (29.7) | 104 | (23.2) | | |
| Post high school | 322 | (50.1) | 87 | (44.6) | 235 | (52.5) | | |
| **Marital Statues** | | | | | | | 30.826[a] | <0.001*** |
| Cohabitation | 392 | (61.0) | 88 | (45.1) | 304 | (68.0) | | |
| Married living alone | 196 | (30.5) | 81 | (44.5) | 115 | (25.7) | | |
| Not married | 54 | (8.4) | 26 | (13.3) | 28 | (6.2) | | |
| **Income to Poverty** | | | | | | | 6.528[a] | 0.011* |
| Impoverished | 186 | (28.9) | 70 | (35.8) | 116 | (25.9) | | |
| Moderate income | 456 | (71.0) | 125 | (64.1) | 331 | (74.0) | | |
| **BMI Status** | | | | | | | −20.140[b] | <0.001*** |
| Underweight | 1 | (0.1) | 1 | (0.5) | 0 | (0.0) | | |
| Normal weight | 84 | (13.0) | 15 | (7.6) | 69 | (15.4) | | |
| Overweight | 190 | (29.5) | 46 | (23.5) | 144 | (32.2) | | |
| Obese | 367 | (57.1) | 133 | (86.2) | 234 | (52.3) | | |

[a] Chi-square test, [b] rank sum test, *P<0.05, **P<0.01, ***P<0.001, same as below.

### 3.2 PAs predictors

Significant statistical differences were observed only in RPA (P<0.001) between the depression group and the non-depression group. Differences were not observed in MVPA (P=0.949), WPA (P=0.203), TPA (P=0.299) and SB (P=0.219). (See Table 2)

   **3.3.1 Association of RPA and diabetes Co-morbid depression.** In analysis for the association of RPA and diabetes co-morbid depression, gender, marital status, income to poverty and BMI status were included in the regression models. Model I (without excluding any confounders) showed an odds ratio (OR) of 0.508 (95% CI: 0.347-0.742) (P<0.001) for the association of RPA and diabetes co-morbid depression; Model II (adjusted for variables of gender) showed OR=0.533 (95% CI: 0.363-0.783) (P=0.001), Model III (adjusted for marital status) showed OR=0.516 (95% CI: 0.678-0.891) (P=0.001); Model IV (adjusted for income to poverty) showed OR=0.530 (95% CI: 0.353-0.774) (P=0.001). Model V (adjusted for BMI status) showed OR=0.522 (95% CI: 0.356-0.789) (P=0.002). The

**Table 2. Physical activity characteristics of NHANES 2017–18 diabetes adults aged ≥20 by depression.**

| Predictor | Sample Capacity | | Depression | | Non-Depression | | OR | 95% CI | P |
|---|---|---|---|---|---|---|---|---|---|
| | N = 642 | | n = 195 | | n = 447 | | | | |
| MVPA | | | | | | | 0.989 | 0.695-1.406 | 0.949 |
| Yes | 416 | (57.1) | 126 | (64.6) | 290 | (64.8) | | | |
| No | 226 | (42.8) | 69 | (35.3) | 157 | (35.1) | | | |
| WPA | | | | | | | 1.233 | 0.807-1.735 | 0.203 |
| Yes | 254 | (39.5) | 84 | (43.0) | 170 | (38.0) | | | |
| No | 388 | (60.4) | 111 | (56.9) | 277 | (61.9) | | | |
| TPA | | | | | | | 1.277 | 0.857-1.904 | 0.299 |
| Yes | 139 | (21.6) | 48 | (24.6) | 91 | (23.0) | | | |
| No | 503 | (78.3) | 147 | (75.3) | 356 | (79.6) | | | |
| **RPA** | | | | | | | **0.508** | **0.347-0.742** | **<0.001***** |
| Yes | 219 | (34.1) | 47 | (24.1) | 172 | (38.4) | | | |
| No | 422 | (65.8) | 148 | (75.8) | 275 | (61.5) | | | |
| SB | | | | | | | 1.298 | 0.856-1.967 | 0.219 |
| Yes | 123 | (19.1) | 43 | (22.0) | 80 | (17.8) | | | |
| No | 519 | (80.8) | 152 | (77.9) | 367 | (82.1) | | | |

results suggest that moderate to vigorous RPA remains a protective factor for depression among diabetic participants after adjusted for confounders. Diabetic participants attending moderate to vigorous RPA are less likely to develop depression. (All *P* < 0.01) (See Table 3)

## 4 Discussion

Through Logistic regression analysis of data from the NHANES 2017−18, we found that among the various types of physical activity, only RPA was a protective factor for co-morbid depression in diabetes. However, although the association of WPA/TPA/SB with diabetes co-morbid depression were not significant, these three predictors may be risk factors for diabetic co-morbid depression in terms of the OR value. In addition, the potential negative effect of WPA/TPA may have contributed to the negative PA results in this study. Therefore, we will discuss "RPA and Diabetes Co-morbid Depression" and "WPA/TPA/SB/MVPA and Diabetes Co-morbid Depression" respectively in the following paragraph.

**Table 3. Results of logistic regression analysis of recreational physical activity and diabetes co-morbid depression from NHANES 2017−18.**

| Mode | b | SE | Wald | P | OR (95% CI) |
|---|---|---|---|---|---|
| I[c] | −0.678 | 0.194 | 12.256 | <0.001*** | 0.508 (0.347-0.742) |
| II[d] | −0.629 | 0.196 | 10.280 | 0.001** | 0.533 (0.363-0.783) |
| III[e] | −0.661 | 0.200 | 10.913 | 0.001** | 0.516 (0.349-0.764) |
| IV[f] | −0.649 | 0.201 | 10.467 | 0.001** | 0.522 (0.353-0.774) |
| V[g] | −0.635 | 0.203 | 9.781 | 0.002** | **0.530 (0.356-0.789)** |

[c] Original model without adjusting for any variables.

[d] Adjusted for the independent variables in Model I plus gender.

[e] Adjusted for variables in Model II plus marital status.

[f] Adjusted for the independent variables in Model III plus income to poverty.

[g] Adjusted for variables in Model IV plus BMI status.

## 4.1 RPA and diabetes Co-morbid depression

RPAs has been found to be a protective factor against co-morbid depression in individuals with diabetes through multiple perspectives. From a social perspective, RPA can improve social support, self-esteem, and a sense of control, which can all positively impact mental health, especially among patients with chronic conditions, as they are more likely to suffer from depression due to psychological burden [23]. A meta analysis based on randomized controlled trials identified from Cochrane review and searches of major electronic databases support the claim that exercise, as a format of RPA, is an good evidence-based treatment for depression [24]. Epidemiological studies suggest that RPA reduces the risk of developing depression in those with diabetes by enhancing insulin sensitivity and glucose uptake, improving cardiovascular fitness and blood pressure regulation, and decreasing inflammation [25,26]. Additionally, due to its recreational function, RPA can increase production of neurotransmitters such as dopamine and endorphins, which have positive effects on mood and cognitive function. Biological mechanisms suggest that regular physical activity can lead to structural and functional brain changes that may be protective against depression. Studies have shown that increased exercise is associated with greater grey matter volume and improved connectivity in brain regions associated with emotional regulation [27,28]. Furthermore, RPA has been linked with changes in neurotrophic factors, which promote neural growth and may have anti-inflammatory and antiapoptotic effects [29]. Researches have shown that the benefits of RPAs on mental health in individuals with diabetes extend to both aerobic and resistance training [30,31], and that even moderate levels of physical activity can have a positive impact on mood.

RPAs can help prevent depression and exert a positive impact on emotional states. However, a study indicates that among numerous exercise programs, yoga and strength training yield better effects [32]. Another randomized controlled trial demonstrates that compared to basic stretching exercises, aerobic exercise is more effective in alleviating depression [33]. Therefore, further research and validation may be required to determine which type of exercise is more effective for depression intervention. Additionally, it is crucial to manage the intensity and duration of exercise appropriately, as excessive physical activity may not necessarily be beneficial [15,34,35].

## 4.2 WPA/TPA/SB/MVPA and diabetes co-morbid depression

The results of the present study demonstrate that MVPA had no significant effect on diabetic co-morbid depression. This differs from the results of previous studies, potentially due to the fact that the physical activity in our study encompassed various types. Not all forms of physical activity are beneficial to health [14,15]; it is possible that the negative impact of work-related physical activity (WPA) on depression offset the positive influence of recreational physical activity on depression. Meanwhile, WPA, TPA, and SB may represent potential risk factors for depression. The negative effect of SB on depression has been widely demonstrated [36], regardless of whether the sample had diabetes or not. Therefore, we mainly focused on the possible reasons for the negative effect of WPA/TPA on diabetes co-morbid depression.

Work or occupational PAs has been identified as a risk factor for co-morbid depression in individuals with diabetes through multiple mechanisms, including stress, psycho-social factors, and biological mechanisms. From a stress perspective, job demands, lack of support or control, and job insecurity can contribute to chronic stress, which has been linked to the development of depression [37]. In terms of psycho-social factors, job strain, work-family conflict, and low social support have also been associated with higher rates of depression among individuals with diabetes [38]. Finally, biological mechanisms suggest that work-related stress can lead to changes in inflammatory cytokines, cortisol levels, and peripheral insulin resistance, which have all been associated with depression and diabetes [39,40]. Researchers have suggested that interventions that improve occupational health and well-being, such as work-site wellness programs, stress-management training, and changes in work schedule, may be promising approaches to reducing co-morbid depression in individuals with diabetes who were still at work which acquires physical movements.

Little research had mentioned why TPA may have a negative effect on diabetic co-morbid depression. Most of the previous studies have shown a preventive alleviative effect of TPA on depression. However, since the metabolic mechanisms of diabetic patients differ from those of normal individuals, which may lead to differences in the effects they obtain when participating in TPA. The exact mechanism of this needs to be further investigated. Finally, this study found there were no significant effect of MVPA on diabetes co-morbid depression, which is different from previous studies. We found that the "MVPA" involved in the previous study was mostly RPAs such as "exercise" and "games", which was different from the comprehensive "MVPA" involved in the present study. RPA had a positive effect on diabetic co-morbid depression, while WPA/TPA may have a negative effect. The effects of these predictors may cancel each other out, ultimately leading to a non-significant effect in MVPA.

### 4.3 Other factors and diabetes co-morbid depression

In addition to physical activity being closely associated with the comorbidity of diabetes and depression, other potential confounding factors may also be relevant. Therefore, we will discuss these other potential factors in the following section.

Gender remains one of the strongest risk factors for depressive symptoms. Compared to men with diabetes, women with diabetes exhibit a significantly higher prevalence of depression [41]. Moreover, older adults are also at a high risk of depression [42]. There is a clear relationship between a deteriorating socioeconomic environment and depression, with higher levels of depressive symptoms being associated with being single, having lower household income, receiving reduced support from friends or family, and experiencing increased stress in the workplace or at home [43,44]. Lifestyle is also closely linked to depression; for instance, a sedentary lifestyle is associated with an increased incidence of depression [45]. Conversely, appropriate physical exercise and moderate alcohol consumption can help prevent depression [46].

All the aforementioned factors may be associated with depression, thereby influencing the research findings. The influencing factors of depression and diabetes are intricately intertwined, involving knowledge and mechanisms across multiple disciplines. There remain numerous unclear aspects that warrant our further exploration.

### 4.4 Limitations of this study

This study has several limitations: 1) General limitations of NHANES cross-sectional studies: including recall bias: In the NHANES cross-sectional study, data were typically collected using self-report measures, and this may lead to recall bias and participants may not accurately report their behavior or condition due to memory errors or other factors; Limited information on duration and intensity: There may be limitations in this study in collecting information on the duration and intensity, such as the severity of diabetes, which may limit the accuracy of findings; Missing data: This study had a high number of missing data (151/893), which may limit the ability to make inferences about the relationships between variables. 2) Environmental-specific limitation: This study extracted data from 2017−18, but the emergence of the COVID-19 pandemic in 2020 raised many issues that directly or indirectly had a greater impact on the mental health of the general population, leading to changes in depression in the present post-epidemic era [22]. In our future design, we will consider avoiding these limitations and further explore the deeper mechanics of PAs, especially WPA and TPA effecting diabetes co-morbid depression based on the innovative points of this study, including not limited to biochemical and genetic studies.

## 5 Conclusions

The results of this study suggested that among various types of physical activity, only RPA was a protective factor for co-morbid depression in diabetes. These findings provide further evidence that 1) not all types of physical activity are beneficial to prevent depression among diabetic patients and 2) physical activity is not equivalent to exercise, as the psychological benefits of exercise may be derived from its recreational function. The findings from this study also provide a valid empirical basis for the prevention methods as well as exercise interventions of diabetic co-morbid depression.

## Acknowledgments

We would like to thank all the staff and participants of the National Health and Nutrition Examination Survey 2017–18 cycles for their valuable contributions. Any interpretation or conclusion related to this manuscript does not represent the views of the CDC or the NHANES. We would also like to thank the editors and reviewers for their valuable and constructive comments to help us improve the manuscript. And we especially thank Dr. JH for his constructive suggestions.

## Author contributions

**Conceptualization:** Yage Yang, Hongzhen Liu.

**Data curation:** Yage Yang.

**Formal analysis:** Yage Yang, Hongzhen Liu.

**Funding acquisition:** Hongzhen Liu.

**Investigation:** Yage Yang, Hongzhen Liu.

**Methodology:** Yage Yang, Hongzhen Liu.

**Project administration:** Hongzhen Liu.

**Software:** Yage Yang.

**Supervision:** Hongzhen Liu.

**Visualization:** Yage Yang.

**Writing – original draft:** Yage Yang, Hongzhen Liu.

**Writing – review & editing:** Hongzhen Liu.

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
