## [Decision Letter · Decision Letter 0]

29 Jul 2025

PONE-D-25-27126The Association of Different Types of Physical Activity and Diabetes Co-morbid Depression: A Cross-sectional AnalysisPLOS ONE

Dear Dr. Liu,

Thank you for submitting your manuscript to PLOS ONE. After careful consideration, we feel that it has merit but does not fully meet PLOS ONE’s publication criteria as it currently stands. Therefore, we invite you to submit a revised version of the manuscript that addresses the points raised during the review process.

Please review the comments provided by the reviwer #1.  It would make a better approach of your paper.  I am going to referee it with another reviewer if possible.

We look forward to receiving your revised manuscript.

Kind regards,

Mynor G. Rodriguez-Hernandez, Ph.D.

Academic Editor

PLOS ONE

Journal Requirements:

https://www.researchsquare.com/article/rs-3095190/v1

https://pmc.ncbi.nlm.nih.gov/articles/PMC10367360/

In your revision ensure you cite all your sources (including your own works), and quote or rephrase any duplicated text outside the methods section. Further consideration is dependent on these concerns being addressed.

3. Please note that your Data Availability Statement is currently missing the accession number of each dataset OR a direct link to access each database. If your manuscript is accepted for publication, you will be asked to provide these details on a very short timeline. We therefore suggest that you provide this information now, though we will not hold up the peer review process if you are unable.

4. In the online submission form, you indicated that [The datasets generated and/or analyzed during the current study are available in the [NHANES] repository, [NHANES Questionnaires, Datasets, and Related Documentation (cdc.gov)]. Raw data supporting the obtained results are available at the corresponding author.].

Additional Editor Comments:

Dear Author, I finally got the reviewers´ comments for your paper. I do apologize for the delay, but it was hard to find them.

Reviewers' comments:

Reviewer's Responses to Questions

**Comments to the Author**

1. Is the manuscript technically sound, and do the data support the conclusions?

Reviewer #1: Partly

Reviewer #2: Yes

2. Has the statistical analysis been performed appropriately and rigorously? 

Reviewer #1: I Don't Know

Reviewer #2: Yes

3. Have the authors made all data underlying the findings in their manuscript fully available?

Reviewer #1: Yes

Reviewer #2: Yes

4. Is the manuscript presented in an intelligible fashion and written in standard English?

Reviewer #1: Yes

Reviewer #2: Yes

5. Review Comments to the Author

Reviewer #1: Thank you for giving me the opportunity to review this manuscript.This study examined the association between various physical activities (MVPA, WPA, TPA, RPA, SB) and co-morbid depression in diabetes using NHANES data. Results showed that only recreational physical activity (RPA) significantly reduced depression risk in diabetes, while other activity types had no significant effect.

- major

1) Please describe the specific recruitment or data collection periods, or the specific locations where participants were surveyed.

2) Please describe how participants were selected and how exclusion criteria were applied.

3) Please describe the criteria for how they were measured and assessed (e.g., duration of activity), and how the data from the PAQ-J were processed and how PA was categorized

4) The statement "RPA has been found to be a protective factor against co-morbid depression" makes a strong claim but fails to address potential confounders or nuances in the literature. Please discuss more critical analysis of varying studies and their findings related to RPA's effectiveness.

5) The study reports a lack of significance for MVPA’s effect on depression but does not fully reconcile this with the existing literature, where MVPA is often linked to positive effects on mental health. Please explain why this study’s findings differ from past research.

6) Please address potential confounders (e.g., other health conditions, socio-economic factors, or lifestyle choices) that could have influenced the results in the discusson.

-minor

1) There are some grammar errors. For example, "PAs has been shown to improve glycemic control..." should be corrected to "PAs have been shown...".

2) Please standardize all citations to the same style (APA, AMA, etc.) and include complete reference details like authors, publication year, article title, and journal name.

3) Please add citations from reliable studies and provide specific data or research results, such as "PA has been shown to improve glycemic control (citation needed).

Reviewer #2: The manuscript titled “The Association of Different Types of Physical Activity and Diabetes Co-morbid Depression: A Cross-sectional Analysis” presents a clear and methodologically rigorous investigation into the relationship between physical activity and depression in individuals with diabetes, using data from the NHANES 2017-18 survey.

1.Technical Soundness and Data Support

The statistical analysis has been performed appropriately. The study utilizes robust methods, including binary logistic regression, to analyze the data. The sample size of 642 participants is adequate for the analysis, and the authors have carefully adjusted for relevant confounders, such as gender, marital status, and BMI. The conclusions drawn are consistent with the data presented, particularly the significant association between recreational physical activity and reduced depression in diabetic individuals, which is well-supported by the statistical evidence.

2.Statistical Analysis

The statistical analyses are rigorous and appropriately applied. The binary logistic regression models used are appropriate for the research question, and the authors have adjusted for multiple potential confounders. The inclusion of different models with various adjustments adds robustness to the findings. There is clear explanation and justification for each statistical procedure.

3.Data Availability

The authors have made the underlying data available. The datasets used in the study are publicly available through the NHANES repository, and the authors have provided sufficient information on how to access the data, adhering to PLOS ONE's data policy. This transparency ensures the reproducibility and verifiability of the results.

4.Presentation and Language

The manuscript is well-organized and written in clear, standard English. The findings are presented logically, and the writing is generally precise and free from grammatical or typographical errors. The language is accessible, and there is no ambiguity in the interpretation of the results.

5.Additional Comments

The study adheres to ethical guidelines, as evidenced by the inclusion of an ethics statement and proper consent from participants. The NHANES study has been approved by the relevant ethics review board, and the authors have followed appropriate ethical standards in the use of data.

There is no indication that the manuscript involves dual publication or has been previously published in part elsewhere. The manuscript appears original and adheres to publication ethics.

In summary, this is a well-conducted study with appropriate statistical analysis, clear presentation, and transparency in data availability. The research provides valuable insights into the potential benefits of recreational physical activity for mental health in individuals with diabetes. I recommend the manuscript for publication with no major concerns.

6. PLOS authors have the option to publish the peer review history of their article (what does this mean?). If published, this will include your full peer review and any attached files.

Reviewer #1: No

Reviewer #2: **Yes: **Zhixin ZHANG

---

## [Author Response · Author response to Decision Letter 1]

23 Aug 2025

We sincerely thank the editor and reviewers for his valuable feedback that we have used to improve the quality of our manuscript. Comments from the reviewer are laid out below in italicized font and specific concerns have been numbered. Our response is given in normal font and changes/additions to the manuscript are given in blue text.

Reviewer #1:

1.Please describe the specific recruitment or data collection periods, or the specific locations where participants were surveyed.

Response: Thank you very much for your valuable suggestions. All of our data were extracted from the NHANES database, and we used data from 2017-2018. NHANES is a population-based cross-sectional survey designed to collect information on the health and nutritional status of the US population. The above content is presented in section 2.1 of the manuscript.

2.Please describe how participants were selected and how exclusion criteria were applied.

Response: We added descriptions of the selection criteria and exclusion criteria. The content is as follows: After excluding missing and invalid data, a final total of 642 samples were included in this study. For more detailed information regarding the study design, sampling, and exclusion criteria, please refer to the figure below. (See Figure 1)

Figure1| Flow chart of subject selection

3.Please describe the criteria for how they were measured and assessed (e.g., duration of activity), and how the data from the PAQ-J were processed and how PA was categorized.

Response: We have rewritten the measurement and evaluation criteria based on previous literature and cited the relevant references. In addition, in this study, we focused primarily on the association between different types of physical activity and comorbid diabetes and depression. In future studies, we will conduct further research on the intensity and duration of physical activity. The revised content is as follows:

This questionnaire classifies physical activity into the following four categories: work-related physical activity, recreational physical activity, commuting physical activity, and sedentary behavior (14). The adult section of this questionnaire encompasses items from PAQ605 to PAQ680. Designed based on the Global Physical Activity Questionnaire (GPAQ), it can provide interview data on various types of physical activity at the respondent level, enabling the assessment of whether the sample engaged in different types of physical activity during a typical week (15).

The references cited are as follows:

Zhang J, Cao Y, Mo H, Feng R. The association between different types of physical activity and smoking behavior. BMC Psychiatry. 2023 Dec 11;23(1):927. doi: 10.1186/s12888-023-05416-1.

Zhang J, Mo H, Feng J, Jiao Z, Yang W, Xue Z, Feng R. Association between physical activity across diversed domains and sedentary behavior with sleep disorders. Sci Rep. 2025 May 7;15(1):15911. doi: 10.1038/s41598-025-00857-y.

4.The statement "RPA has been found to be a protective factor against co-morbid depression" makes a strong claim but fails to address potential confounders or nuances in the literature. Please discuss more critical analysis of varying studies and their findings related to RPA's effectiveness.

Response: We have enriched the discussion section by incorporating diverse research findings and have extensively read and referenced additional literature. The content is as follows: RPAs can help prevent depression and exert a positive impact on emotional states. However, a study indicates that among numerous exercise programs, yoga and strength training yield better effects (32). Another randomized controlled trial demonstrates that compared to basic stretching exercises, aerobic exercise is more effective in alleviating depression (33). Therefore, further research and validation may be required to determine which type of exercise is more effective for depression intervention. Additionally, it is crucial to manage the intensity and duration of exercise appropriately, as excessive physical activity may not necessarily be beneficial (15, 34, 35).

5.The study reports a lack of significance for MVPA’s effect on depression but does not fully reconcile this with the existing literature, where MVPA is often linked to positive effects on mental health. Please explain why this study’s findings differ from past research.

Response: We have added explanations to the manuscript regarding the differences between the results of this study and those of previous studies. The content is as follows: The results of the present study demonstrate that MVPA had no significant effect on diabetic co-morbid depression. This differs from the results of previous studies, potentially due to the fact that the physical activity in our study encompassed various types. Not all forms of physical activity are beneficial to health (14,15); it is possible that the negative impact of work-related physical activity (WPA) on depression offset the positive influence of recreational physical activity on depression.

6.Please address potential confounders (e.g., other health conditions, socio-economic factors, or lifestyle choices) that could have influenced the results in the discusson.

Response: We have incorporated discussions on other relevant factors. The content is as follows:

4.3 Other Factors and Diabetes Co-morbid Depression

In addition to physical activity being closely associated with the comorbidity of diabetes and depression, other potential confounding factors may also be relevant. Therefore, we will discuss these other potential factors in the following section.

Gender remains one of the strongest risk factors for depressive symptoms. Compared to men with diabetes, women with diabetes exhibit a significantly higher prevalence of depression (42). Moreover, older adults are also at a high risk of depression (43). There is a clear relationship between a deteriorating socioeconomic environment and depression, with higher levels of depressive symptoms being associated with being single, having lower household income, receiving reduced support from friends or family, and experiencing increased stress in the workplace or at home (44, 45). Lifestyle is also closely linked to depression; for instance, a sedentary lifestyle is associated with an increased incidence of depression (46). Conversely, appropriate physical exercise and moderate alcohol consumption can help prevent depression (47).

All the aforementioned factors may be associated with depression, thereby influencing the research findings. The influencing factors of depression and diabetes are intricately intertwined, involving knowledge and mechanisms across multiple disciplines. There remain numerous unclear aspects that warrant our further exploration.

7.There are some grammar errors. For example, "PAs has been shown to improve glycemic control..." should be corrected to "PAs have been shown...".

Response: Thank you very much for pointing out our mistake. We have changed "PAs has been shown to improve glycemic control..." to "PAs have been shown...".

8.Please standardize all citations to the same style (APA, AMA, etc.) and include complete reference details like authors, publication year, article title, and journal name.

Response: We have standardized the citation format and listed the reference information in full, including author names, publication year, article title, and journal name.

9.Please add citations from reliable studies and provide specific data or research results, such as "PA has been shown to improve glycemic control (citation needed).

Response: Thank you very much for your valuable suggestions. We have cited the relevant references in the manuscript and described the relevant research results. The content is as follows: PAs have been shown to improve glycemic control, insulin sensitivity, blood pressure, and lipid profiles, as well as have positive effects on mood and psychological well-beings (2). A meta-analysis of 32 randomized controlled trials showed that PA interventions were effective in reducing depressive symptoms among adults with diabetes (3). Furthermore, a longitudinal cohort study also suggested that higher levels of physical activity were associated with a reduced risk of developing depression in individuals with type II diabetes (4).

The references cited are as follows:

Warburton DER, Bredin SSD. Health benefits of physical activity: a systematic review of current systematic reviews. Curr Opin Cardiol. 2017 Sep;32(5):541-556. doi: 10.1097/HCO.0000000000000437.

van der Feltz-Cornelis C, Allen SF, Holt RIG, Roberts R, Nouwen A, Sartorius N. Treatment for comorbid depressive disorder or subthreshold depression in diabetes mellitus: Systematic review and meta-analysis. Brain Behav. 2021 Feb;11(2):e01981. doi: 10.1002/brb3.1981.

Palakodeti S, Uratsu CS, Schmittdiel JA, Grant RW. Changes in physical activity among adults with diabetes: a longitudinal cohort study of inactive patients with Type 2 diabetes who become physically active. Diabet Med. 2015 Aug;32(8):1051-7. doi: 10.1111/dme.12748.

Reviewer #2:

The manuscript titled “The Association of Different Types of Physical Activity and Diabetes Co-morbid Depression: A Cross-sectional Analysis” presents a clear and methodologically rigorous investigation into the relationship between physical activity and depression in individuals with diabetes, using data from the NHANES 2017-18 survey.

1.Technical Soundness and Data Support

The statistical analysis has been performed appropriately. The study utilizes robust methods, including binary logistic regression, to analyze the data. The sample size of 642 participants is adequate for the analysis, and the authors have carefully adjusted for relevant confounders, such as gender, marital status, and BMI. The conclusions drawn are consistent with the data presented, particularly the significant association between recreational physical activity and reduced depression in diabetic individuals, which is well-supported by the statistical evidence.

2.Statistical Analysis

The statistical analyses are rigorous and appropriately applied. The binary logistic regression models used are appropriate for the research question, and the authors have adjusted for multiple potential confounders. The inclusion of different models with various adjustments adds robustness to the findings. There is clear explanation and justification for each statistical procedure.

3.Data Availability

The authors have made the underlying data available. The datasets used in the study are publicly available through the NHANES repository, and the authors have provided sufficient information on how to access the data, adhering to PLOS ONE's data policy. This transparency ensures the reproducibility and verifiability of the results.

4.Presentation and Language

The manuscript is well-organized and written in clear, standard English. The findings are presented logically, and the writing is generally precise and free from grammatical or typographical errors. The language is accessible, and there is no ambiguity in the interpretation of the results.

5.Additional Comments

The study adheres to ethical guidelines, as evidenced by the inclusion of an ethics statement and proper consent from participants. The NHANES study has been approved by the relevant ethics review board, and the authors have followed appropriate ethical standards in the use of data.

There is no indication that the manuscript involves dual publication or has been previously published in part elsewhere. The manuscript appears original and adheres to publication ethics.

In summary, this is a well-conducted study with appropriate statistical analysis, clear presentation, and transparency in data availability. The research provides valuable insights into the potential benefits of recreational physical activity for mental health in individuals with diabetes. I recommend the manuscript for publication with no major concerns.

Response: We're truly delighted and grateful for your positive feedback and recognition of our paper. Your endorsement means a lot to us and validates our efforts. We're encouraged by your approval and will continue to strive for excellence in our future research. Thank you again for your time and valuable insights!

---

## [Editor Report · Decision Letter 1]

3 Sep 2025

The Association of Different Types of Physical Activity and Diabetes Co-morbid Depression: A Cross-sectional Analysis

PONE-D-25-27126R1

Dear Dr .Hongzhen Liu

We’re pleased to inform you that your manuscript has been judged scientifically suitable for publication and will be formally accepted for publication once it meets all outstanding technical requirements.

Kind regards,

Mynor G. Rodriguez-Hernandez, Ph.D.

Academic Editor

PLOS ONE

Additional Editor Comments (optional):

I do apologize for the delay on this decision, but I was waiting on previous reviewers' comments.

Please review the conclusion statement and make sure it is the take-home message you want to share. Right now sounds more like a result. The abstract could be improved by providing a clear statement of what the actual conclusion/result means in terms of health and quality of life for this specific population.
---

## [Editor Report · Acceptance letter]

PONE-D-25-27126R1

PLOS ONE

Dear Dr. Liu,

I'm pleased to inform you that your manuscript has been deemed suitable for publication in PLOS ONE. Congratulations! Your manuscript is now being handed over to our production team.

Kind regards,

on behalf of

Dr. Mynor G. Rodriguez-Hernandez

Academic Editor

PLOS ONE